# Combining Fractional Derivatives and Machine Learning: A Review

**DOI:** 10.3390/e25010035

**Published:** 2022-12-24

**Authors:** Sebastian Raubitzek, Kevin Mallinger, Thomas Neubauer

**Affiliations:** 1Data Science Research Unit, TU Wien, Favoritenstrasse 9-11/194, 1040 Vienna, Austria; 2SBA Research gGmbh, Floragasse 7, 1040 Vienna, Austria

**Keywords:** fractional derivative, fractional calculus, machine learning, artificial intelligence, complexity, regression analysis

## Abstract

Fractional calculus has gained a lot of attention in the last couple of years. Researchers have discovered that processes in various fields follow fractional dynamics rather than ordinary integer-ordered dynamics, meaning that the corresponding differential equations feature non-integer valued derivatives. There are several arguments for why this is the case, one of which is that fractional derivatives inherit spatiotemporal memory and/or the ability to express complex naturally occurring phenomena. Another popular topic nowadays is machine learning, i.e., learning behavior and patterns from historical data. In our ever-changing world with ever-increasing amounts of data, machine learning is a powerful tool for data analysis, problem-solving, modeling, and prediction. It has provided many further insights and discoveries in various scientific disciplines. As these two modern-day topics hold a lot of potential for combined approaches in terms of describing complex dynamics, this article review combines approaches from fractional derivatives and machine learning from the past, puts them into context, and thus provides a list of possible combined approaches and the corresponding techniques. Note, however, that this article does not deal with neural networks, as there is already extensive literature on neural networks and fractional calculus. We sorted past combined approaches from the literature into three categories, i.e., preprocessing, machine learning and fractional dynamics, and optimization. The contributions of fractional derivatives to machine learning are manifold as they provide powerful preprocessing and feature augmentation techniques, can improve physically informed machine learning, and are capable of improving hyperparameter optimization. Thus, this article serves to motivate researchers dealing with data-based problems, to be specific machine learning practitioners, to adopt new tools, and enhance their existing approaches.

## 1. Introduction

Machine learning and fractional calculus are tools capable of dealing with and describing complex real-life phenomena and its relation to its inherent nonlinear properties.

Whereas machine learning dynamically learns complex behavior from data in most cases, the framework of fractional calculus was previously used to describe complex phenomena by statically modeling them. The issue of a fractional derivative first came up in a letter from de l’Hopital to Leibniz in 1695, i.e., what one would obtain from a derivative of non-integer order *n*, e.g., $n=\frac{1}{2}$. Leibniz’s famous response to this day was: “It will lead to a paradox from which one day useful consequences will be drawn” [1].

Nowadays, both frameworks feature a multitude of applications in various disciplines.

The applications of fractional calculus range from physics, wherein one can describe anomalous diffusion in complex liquids or frequency-dependent acoustic wave propagation, to, e.g., image processing, where one can use fractional derivatives as filters or to enhance images [2]. Another area of research in which fractional calculus is used to describe various phenomena is that of the environmental sciences. Here, fractional calculus is applied to interpret hydrological cycles or analyze underground water chemical reactions [2,3]. As discussed by Du et al. [4], fractional calculus introduces memory into modeling processes, which in turn is required for specific approaches in, e.g., biology or psychology. Thus, the researchers suggested interpreting the fractional order of a process as a degree of memory of this process [4].

This article aimed to find overlaps between machine learning and fractional calculus, specifically fractional derivatives. Furthermore, we sought to discuss these overlaps or combined applications and put them into a bigger context and finally, give some recommendations on using fractional derivatives to enhance machine learning approaches. At this point, we need to clarify that, when employing the popular term machine learning and narrowing down the scope of this article, we only talk about supervised non-neural-network machine learning applications. The reason for not including neural networks is that there already is an excellent article on the combined approaches of neural networks and fractional calculus. Thus, we are not going to discuss this topic again but instead, refer the reader to the article by Viera et al. [5]. Still, we want to highlight two articles from this review to motivate the topic. Khan et al. [6], presented a fractional backpropagation algorithm for training neural networks. Furthermore, the work in Ref. [7] showed the applicability of a particle swarm optimization algorithm with a fractional order neural network.

We also want to clarify that the scope of this article is to provide additional tools from fractional calculus to the experienced machine learning practitioner rather than introducing people familiar with fractional calculus to the realm of machine learning. Thus, the outcome of this article is a focused list of possible combined applications of machine learning and fractional derivatives and, based on the discussed literature, a discussion on the context of the combination of these two tools as well as recommendations for future research.

This article is structured as follows:

Section 2 describes how this review was performed, the employed search, and the exclusion criteria. Section 3 briefly introduces fractional calculus and describes the different types of fractional derivatives. Section 4 introduces supervised machine learning. Then, we present the results of our literature review in Section 5. We discuss our findings and put them into context in Section 6 and conclude this article in Section 7. We further added Appendix A, which provides a table containing all reviewed articles.

## 2. Methodology

For this review, we performed an online search on (google scholar) https://scholar.google.com/. We searched for the keywords listed in the following list:
Machine learning fractional calculusMachine learning fractional derivativesSupport vector fractional calculusSupport vector fractional derivativesDecision tree fractional calculusDecision tree fractional derivativeRandom forest fractional calculusRandom forest fractional derivativeXGBoost fractional calculusXGBoost fractional derivativeRegression fractional calculusRegression fractional derivativeRidge regression fractional calculusRidge regression fractional derivativeLasso regression fractional calculusLasso regression fractional derivativeLogistic regression fractional calculusLogistic regression fractional derivativeNaive Bayes fractional calculusNaive Bayes fractional derivativeKNN fractional calculusKNN fractional derivativeNearest neighbor fractional calculusNearest neighbor fractional derivativeLightGBM fractional calculusLightGBM fractional derivativeExtreme learning fractional calculusExtreme learning fractional derivativeExtreme boosting fractional calculusExtreme boosting fractional derivative

We acknowledge that this list is incomplete; further, it can never be a complete list. This is because of the somewhat recently popularized blurry use of the term machine learning and lingo and the ever-changing landscape of new algorithms and ideas in the field. Here, we want to refer to two homepages [8,9], which are both glossaries that keep up with the fast-changing machine learning terminology. Thus, the previous list contains popular machine learning algorithms, prevalent machine learning, and fractional calculus keywords, and occasionally some more niche keywords, which are included because of the authors’ domain knowledge.

Furthermore, the authors added some articles that did not appear in the mentioned search queries but were deemed to be a relevant addition to this article based on the authors’ domain knowledge.

As we usually found articles fitting our exclusion criteria within the first ten results, we prioritized our search and analysis to the first 20 results for each search query.

As already mentioned, we employed a range of exclusion criteria to keep this review focused and to reduce the number of featured articles. Thus, we used the following exclusion criteria:We excluded articles primarily targeting and/or only targeting neural networks.This was done because there already exists an excellent review of neural networks and fractional calculus [5]. Nonetheless, we will peripherally mention neural network applications or feature articles that include neural networks along with other machine learning algorithms.We excluded articles dealing with control problems.This criterion was employed to keep this review focused, as including control problems would have unnecessarily blown up the discussed methodology and made it harder to identify key takeaways for hybrid applications of fractional calculus and machine learning.We discarded everything related to grey models.Grey models do not coincide with the classic supervised learning approaches we are looking for in this review.We excluded publications that dealt with fractional integrals rather than with fractional derivatives.We excluded publications that featured fractional complexity measures, e.g., fractional entropy measures.Nonetheless, we briefly mention them in our discussion if they provide valuable insights into hybrid applications.We excluded unsupervised and reinforcement learning approaches.Thus, we are only dealing with supervised learning.

## 3. Fractional Derivatives

Fractional derivatives are a generalization of integer-ordered derivatives such that they allow for derivatives of the order of $n=\frac{1}{2}$ instead of only integer-valued derivatives, i.e., n=1,2,3,⋯.

Contrary to integer order derivatives, they do not behave locally, i.e., the fractional derivative of a function f(t) depends on all values of the same function, thus inducing memory about every point of a signal into the derivative.

Furthermore, there are many ways to define a fractional derivative; thus, many researchers proposed their version of the fractional derivative, sometimes explicitly tailored to match a particular problem/application. To further illustrate this idea and to give the reader a grasp on the multitude of possible fractional derivatives, we give the list from https://en.wikipedia.org/wiki/Fractional_calculus (Wikipedia) [10]: the listed fractional derivatives are the Grünwald–Letnikov derivative, the Sonin–Letnikov derivative, the Liouville derivative, the Caputo derivative, the Hadamard derivative, the Marchaud derivative, the Riesz derivative, the Miller–Ross derivative, the Weyl derivative, the Erdélyi–Kober derivative, the Coimbra derivative, the Katugampola derivative, the Hilfer derivative, the Davidson derivative, the Chen derivative, the Caputo-Fabrizio derivative, and the Atangana–Baleanu derivative.

Because of this arbitrariness of choosing a fractional derivative, we stick to those fractional derivatives relevant in our literature review, briefly explain them, and reference sources for a discussion on them. Thus, the discussed fractional derivatives are the Grünwald–Letnikov, the Reisz, the Caputo, and the Riemann–Liouville fractional derivatives. Furthermore, we need to differentiate between left- and right-sided derivatives for many of these fractional derivatives. The following list is based on the reviews by de Oliveira et al. and Aslan [11,12], which we refer the reader to for an in-depth discussion of the topic. Furthermore, for a discussion on the discrete version of the fractional derivatives, we refer to [13,14]. The featured applications combining fractional derivatives and machine learning use the following list of fractional derivatives:**The Grünwald–Letnikov fractional derivative**(1)$$\begin{array}{c} \begin{array}{cc} \mathrm{left}\text{-}\mathrm{sided}:D{\left[f\left(x\right)\right]}_{a^{+}}^{\alpha } & =\lim_{h\to 0}\frac{1}{h^{\alpha }}\sum_{k=0}^{\left\lfloor n\right\rfloor }{\left(-1\right)}^{k}\frac{\Gamma \left(\alpha +1\right)f\left(x-kh\right)}{\Gamma \left(k+1\right)\Gamma \left(\alpha -k+1\right)},\;nh=x-a\\ \mathrm{right}\text{-}\mathrm{sided}:D{\left[f\left(x\right)\right]}_{b^{-}}^{\alpha } & =\lim_{h\to 0}\frac{1}{h^{\alpha }}\sum_{k=0}^{\left\lfloor n\right\rfloor }{\left(-1\right)}^{k}\frac{\Gamma \left(\alpha +1\right)f\left(x-kh\right)}{\Gamma \left(k+1\right)\Gamma \left(\alpha -k+1\right)},\;nh=b-x \end{array} \end{array}$$**The Caputo Fractional Derivative**(2)$$\begin{array}{c} \begin{array}{cc} \mathrm{left}\text{-}\mathrm{sided}:D_{a^{+}}^{\alpha }\left[f\left(x\right)\right] & =\frac{1}{\Gamma \left(n-\alpha \right)}\int_{a}^{x}{\left(x-\zeta \right)}^{n-\alpha -1}\frac{d^{n}}{d\zeta^{n}}\left[f\left(\zeta \right)\right]d\zeta ,\;x\geq a\\ \mathrm{right}\text{-}\mathrm{sided}:D_{b^{-}}^{\alpha }\left[f\left(x\right)\right] & =\frac{1}{\Gamma \left(n-\alpha \right)}\int_{x}^{b}{\left(x-\zeta \right)}^{n-\alpha -1}\frac{d^{n}}{d\zeta^{n}}\left[f\left(\zeta \right)\right]d\zeta ,\;x\leq b \end{array} \end{array}$$**The Riemann–Liouville fractional derivative**(3)$$\begin{array}{c} \begin{array}{cc} \mathrm{left}\text{-}\mathrm{sided}:D_{a^{+}}^{\alpha }\left[f\left(x\right)\right] & =\frac{1}{\Gamma \left(n-\alpha \right)}\frac{d^{n}}{dx^{n}}\int_{a}^{x}{\left(x-\zeta \right)}^{n-\alpha -1}f\left(\zeta \right)d\zeta ,\;x\geq a\\ \mathrm{right}\text{-}\mathrm{sided}:D_{b^{-}}^{\alpha }\left[f\left(x\right)\right] & =\frac{1}{\Gamma \left(n-\alpha \right)}\frac{d^{n}}{dx^{n}}\int_{x}^{b}{\left(x-\zeta \right)}^{n-\alpha -1}f\left(\zeta \right)d\zeta ,\;x\leq b \end{array} \end{array}$$**The Riesz Fractional Derivative**(4)$$D_{x}^{\alpha }\left[f\left(x\right)\right]=-\frac{1}{2\cos\left(\frac{\alpha \pi }{2}\right)}\frac{1}{\Gamma \left(\alpha \right)}\frac{d^{n}}{dx^{n}}\left(\int_{-\infty }^{x}{\left(x-\zeta \right)}^{n-\alpha -1}f\left(\zeta \right)d\zeta +\int_{x}^{\infty }{\left(\zeta -x\right)}^{n-\alpha -1}f\left(\zeta \right)d\zeta \right)$$

**Remark 1.** 

*α denotes the order of the fractional derivative. Whereas $\alpha \in C:R\in \left(n-1,n\right],\;n\in N$. Thus, n defines the integer proximity of α. $R\left(\;\cdot \;\right)$ denotes the real part of a complex number. Furthermore, as defined here, alpha can be complex. However, every single reviewed article (see Appendix A for the summary table) featuring a combined application of fractional derivatives and machine learning uses a real or at least rational value for α.*

*n−1 and n as $n-1\leq R\left(\alpha \right)<n$.*

*$\left[a,b\right]$ is a finite interval in R and k∈N. Furthermore, $f\left(0\right)\equiv f\left(0^{+}\right)-f\left(0^{-}\right)$.*

*ζ is an auxiliary variable used for integration.*

*$\Gamma \left(\;\cdot \;\right)$ is the gamma function [15].*

*⌊·⌋ denotes the floor function, i.e., ⌊x⌋max{z∈Z:z≤x}*

*The previously mentioned "memory" is induced by the summation over k in the case of the Grünwald–Letnikov derivative and by the integration over ζ for the other mentioned derivatives.*



## 4. Supervised Machine Learning

Supervised machine learning is the practice of using a set of input variables, such as blood metabolite, gene expression, or environmental conditions, to predict a quantitative output, such as diseases in humans or crop yields [16]. Thus, aiming to categorize data from prior information via an algorithm trained from labeled data [17,18].

There are numerous supervised learning algorithms, such as linear regression, support vector machines, decision trees, and random forests, to name a few.

One can further partition supervised machine learning into classification and regression tasks. In classification tasks, the algorithm predicts categorical variables such as types of diseases. In contrast, in regression, the algorithm predicts continuous numerical outputs such as wheat yields in, e.g., Hg/Ha.

We will not further discuss supervised learning for several reasons. First, this review aimed to educate machine learning practitioners about the possible applications of fractional derivatives, thus assuming that the reader is somehow familiar with supervised learning. Second, supervised learning and machine learning, in general, constitute a popular multidisciplinary area of research nowadays. Therefore, it suffers from a fast-changing and ever-increasing landscape of algorithms, terminology, and approaches. Third, there is a vast amount of literature and tutorials available that we cannot cover here. Nonetheless, we want to recommend the books by Jason Brownlee for a tutorial-heavy introduction to the topic [19,20,21].

## 5. Results: Combined Approaches of Fractional Derivatives and Machine Learning

Given the previously defined methodology (Section 2), we found 60 relevant publications. We then separated this list of publications in this section into three main categories on how to combine machine learning and fractional derivatives, i.e., preprocessing, machine learning and fractional modeling and optimization. In the following, we will elaborate on these categories separately and further divide them into subcategories wherever necessary.

### 5.1. Preprocessing

In this section, we discuss the applications of fractional derivatives for data preprocessing in machine learning applications. Part of this discussion is data pre-treatment in general, additional features based on fractional derivatives, and feature extraction based on fractional derivatives. These ideas, though distinct, often overlap, especially when discussing additional features and feature extraction. All these techniques have in common that they are applied before the actual machine learning, e.g., the training process. Thus, we named this Section *preprocessing*.

We sort relevant publications by their type of application, e.g., fractional derivatives used for preprocessing in spectroscopy. In this process, we focus on three categories of applications, i.e., spectroscopy, biomedical and engineering applications.

#### 5.1.1. Spectroscopy

An important area of research for the combined approaches of fractional derivatives and machine learning is spectroscopy. Here, fractional derivatives are used as a preprocessing step to enhance the spectral data and thus, improve the accuracy of the machine learning algorithm. We can again differ between two main categories, i.e., visual and near-infrared (Vis-NIR) spectroscopy and hyperspectral analysis.

Regarding visual and near-infrared spectroscopy, we find a range of agricultural applications. In [22], the researchers estimate organic matter content in arid soil from Vis-NIR spectral data. Here, the Grünwald–Letnikov fractional derivative is used as a pretreatment for the spectral data, i.e., the fractional derivative is applied to the obtained spectral data to improve the accuracy of the employed machine learning algorithms, random forest, and partial least squares regression. Similar to this approach is the approach in [23], where fractional derivatives are again used as a pretreatment to augment the Vis-NIR data to estimate the soil organic content. Here, additionally to the previously used algorithms, the researchers also used memory-based learning.

The same idea, i.e., to augment Vis-NIR data using the G-L derivative to improve machine learning estimates, is used in both [24,25] to estimate the nitrogen content of rubber tree cultivations and in [26] for cotton farming. The employed machine learning algorithms range from partial least squares regression, extreme learning machines, convolutional neural networks, and support vector machines.

For another application in agriculture, Bhadra et al. employed the Grünwald–Letnikov fractional derivative to augment Vis-NIR data to quantify the leaf chlorophyll in sorghum using various machine learning algorithms such as, again, partial least squares regression, random forest, support vector machines, and extreme learning machines [27].

Urbanization and industrialization are putting a lot of stress on the environment and further polluting agricultural land with, e.g., various metals. Here [28,29,30] provided ideas to identify zinc lead and other heavy metals from Vis-NIR spectra by, again, employing data augmentation based on the Grünwald–Letnikov fractional derivative to improve the accuracy of machine learning algorithms such as random forest, XGBoost, extreme learning machines, support vector machines, and partial least squares regression.

Finally, the Grünwald–Letnikov fractional derivative is also used to improve the discussed Vis-NIR data to improve the estimation of soil salinity, soil salt, and other water-soluble ions in [31,32]. Here, again, the researchers used the Grünwald–Letnikov fractional derivative to augment the spectral data and/or obtain spectral indices. Again, various ML algorithms were employed, such as partial least squares regression, random forests, and extreme learning machines.

When it comes to hyperspectral data, we find similar approaches to the discussed Vis-NIR spectral applications, i.e., the spectral data were preprocessed using the Grünwald–Letnikov fractional derivative and afterward a measured observable such as, e.g., soil organic matter is predicted using a machine learning algorithm such as, e.g., partial least squares regression [33]. In [34], estimates on top soil organic carbon were given using a Grünwald–Letnikov fractional derivative and random forest.

Furthermore, the (soil) salt content can be estimated using hyperspectral data and a range of ML algorithms, as performed in [31,35,36]. Another agricultural application is to assess the nitrogen content in apple tree canopies via support vector machines, random forest algorithms, and fractional-derivative-augmented hyperspectral data [37].

Cheng et al. predicted the photosynthetic pigments in apple leaves using fractional derivatives to augment the data and range of machine learning algorithms, i.e., support vector machines, k-nearest neighbor, random forest, and neural networks [38]. In [39], it is shown that a similar approach using a different algorithm, XGBoost, can be used to estimate the moisture content of agricultural soil. Furthermore, in [40], XGBoost, LightGBM, random forests, and other algorithms were used to predict the soil electrical conductivity from hyperspectral data using fractional-derivative-augmented data. Although related, but not necessarily an agricultural application, the use of augmented hyperspectral data to estimate the clay content in desert soils using partial least squares regression was discussed in [41].

Furthermore, these ideas of using fractional derivatives to preprocess hyperspectral data to improve ML approaches are not restricted to only using the remotely sensed data of soils. These ideas were applied to water in both [42,43]. In [42], fractional derivatives were used to improve the prediction of total nitrogen in water using XGBoost and random forest, whereas in [43], a similar approach was used for estimating the total suspended matter in water using random forest.

#### 5.1.2. Biomedical Applications

We also find various applications when it comes to the biomedical applications of fractional derivatives combined with ML.

Starting with the classification of EEG signals, we find three publications that combine machine learning and fractional derivatives. Ref. [44] proposed to the classification of electroencephalography (EEG) signals into ictal and seizure-free signals by employing a fractional linear prediction from a fractional order system to model the signals, and/or parts of the signal, and to then use the thus-obtained parameters as features for a support vector machine classification approach. Similar to this is the approach taken by AAruni et al. [45], wherein EEG signals were modeled using the transfer function of a fractional order system. In contrast to the previous publication, the researchers used the error and signal energy as features for support vector machines to differentiate between ictal and seizure-free signals. The third publication dealing with this subject, i.e., modeling EEG signals using fractional transfer functions and classifying between different categories of EEG signals, e.g., seizure and seizure-free, is [46]. Here, similarly to the two previously referenced publications, the fractional modeling transfer function was used to obtain the error and signal energy to be fed into a k-nearest neighbor classifier.

Furthermore, similar approaches can be used to analyze electrocardiography (ECG) signals. Again, in [47], a fractional order transfer function was applied to the QRS complex signal (from ECG measurements) to obtain six parameters which are then fed into a k-nearest neighbor classifier for person identification. Ref. [48] described an approach to do the same to obtain five parameters from a QRS complex signal to differentiate between healthy and three types of arrhythmic signals using a k-nearest neighbor classifier.

Given the results shown in [49], one can use the same approach, e.g., fractional transfer functions in combination with a k-nearest neighbor classifier, to analyze the respiratory properties of humans. In this process, three model parameters are used as features to classify respiratory diseases.

In [50], the researchers used the Grünwald–Letnikov fractional derivative to preprocess online handwriting data. Afterward, these preprocessed data were fed into a support vector machine and random forest algorithms to detect/classify Parkinson’s disease from a person’s handwriting.

In contrast to all previously listed publications in this section, in [51], a discrete fractional mask was used to preprocess the CT images and to improve the identification of tumors using support vector machines, random forest, J48, and simple cart machine learning models.

#### 5.1.3. Engineering

Apart from the two previously discussed categories, we found three publications that did not fit into spectroscopical or biomedical applications. Instead, we consider these publications to be engineering approaches.

In [52], the researchers used histogram peak distributions and the Grünwald–Letnikov fractional derivative to preprocess the images of solar panels to then identify defective solder joints via extreme learning machines.

Ref. [53] used the Reisz fractional derivative to preprocess spectral data obtained from a fiber Bragg grating sensor. Afterwards, these data were fed into three machine learning algorithms, i.e., random forest, linear regression, decision trees, and a multi-layer perceptron, to detect the temperature peaks on solar panels from the FBG spectrum.

In [54], the researchers identified construction equipment activity from image/video data using a combination of a fractional derivative and a range of machine learning algorithms. The employed machine learning algorithms were random forest, neural networks (a multi-layer perceptron), and support vector machines. The fractional derivative employed was the Riemann–Liouville fractional derivative, and it was used to preprocess the images, which were fed into the algorithms at a later step.

### 5.2. Machine Learning and Fractional Dynamics

In this section, we discuss publications where fractional dynamics and the corresponding modeling via fractional differential equations has been used in combination with machine learning. There are several ways to do this, e.g., fractional modeling and after employing machine learning for a regression or classification using features obtained from the modeling approach or using machine learning to improve the identification of fractional models/equations from data.

Although the publications by [44,45,46,47,48,49] have already been categorized as preprocessing, they also serve as a hybrid machine learning and fractional dynamics approach. Specifically, as the authors employed the transfer function from a fractional order model, first a a signal was modeled to obtain the functions parameters or the corresponding error and signal energy of the model as features for classification.

When identifying the correct transfer function and the underlying dynamics, i.e., with the corresponding fractional differential operators, we found an excellent exemplary application in [55]. Here, the employed machine learning technique was a Gaussian process regression, whereas the unknown equation parameters are treated as the hyperparameters of a physics-informed Gaussian process kernel. This framework was used to identify a linear space-fractional differential equations from data and its applicability is shown for stock market data, e.g., S&P 500 stock data.

In [56], similar ideas were employed to model general damping systems. Here, k-means was employed to first differ between viscous and hysteretic damping. Next, the viscous damping dynamics were recovered using a linear regression approach, and afterwards, the hysteretic damping was identified using a sparse regression approach. This framework also allows for fractional-order models, and its applicability was shown for, e.g., a fractional-order viscoelastic damper.

We further found an application for machine learning to identify fractional-order population growth models from data, as performed in [57]. The researchers developed a least squares support vector machine with an orthogonal kernel for simulating this equation. This orthogonal kernel consists of fractional rational Legendre functions.

In [58], the researchers employed ridge regression as a numerical boundary treatment for the fractional KdV-Burgers equation. Another application for identifying fractional models was given by [59], where the researchers used a Gaussian process regression to identify a fractional chaotic model in finance from data. Furthermore, as the extrapolation of the thus-identified model does not perform well, the researchers suggested employing a neural network to improve prediction accuracy.

In [60], both a fractional order model and a first-order resistor–capacitor model were employed to describe the electrical behavior of battery cells with external short circuit faults. Afterwards, a random forest classifier was fed with features from the so-obtained fractional order model to classify whether a battery cell incurs leakage.

### 5.3. Optimization

In this section, we differ between derivative-free and gradient-based methods within the optimization context. This is also to say that the derivative-free algorithms, e.g., particle swarm algorithms, still employ fractional derivatives in the form of, e.g., fractional velocities. The featured derivative-free algorithms include bio-inspired algorithms such as, e.g., the fractional bee colony algorithm and the fractional particle swarm algorithm.

#### 5.3.1. Fractional Gradient-Based Optimization

When discussing the optimization, the first big topic is gradient-based algorithms. Here, the basic idea is to alter the gradient-based approach by switching to a fractional gradient. One benefit of doing so is that these algorithms do not fall into the trap of the local minimum problem as much as regular gradient-based algorithms (find a good citation here).

Regarding the actual applications where fractional derivatives are used for optimization in combination with machine learning algorithms, we find a range of fractional gradient descent applications. We want to point out that all the discussed combined approaches employ the Caputo-fractional derivative but use different algorithms and datasets. In [61], the researchers used ridge regression in combination with fractional gradient descent on the testing environment and datasets provided by [62]. The same testing environment is used in [63]. However, in this publication, the researchers used logistic regression instead of ridge regression. In both cases, the researchers reported improvements to basic (non-fractional) implementations.

Two more applications of fractional gradient descent are found in [64,65]. In these publications, the researchers combined fractional gradient descent with support vector machines for two classic datasets, i.e., the Iris and the Rainfall datasets.

Contrary to the previous modeling approaches are the ideas presented in [66]; here, a multilinear regression is modified using a fractional derivative based on the input features to the linear regression. Thus, obtaining the derivative and, consequently, the fractional derivative from the minimum of the slope means setting it to zero. This idea is used to predict the gross domestic product in Romania. It is shown that models not considering fractional derivatives can be outperformed using the presented concepts. Awadalla et al. [67] presented a similar approach where the same idea is used for fractional linear and fractional squared regression, i.e., the derivatives to derive the minimum are fractionalized. This approach also shows the applicability of the proposed ideas to standard datasets.

#### 5.3.2. Fractional Gradient-Free Optimization

Particle swarm optimization algorithms are some of the most popular gradient-free optimization techniques. In this section, we further differed between regular and Darwinian particle swarm algorithms. All particle swarm algorithms feature a velocity component, which is an attempt to make the algorithms fractional, is changed to a non-integer derivative-based velocity [68].

Apart from gradient-based algorithms, we find a range of bio-inspired gradient-free algorithms. The following discussion includes a fractional particle swarm, fractional Darwinian particle swarm, and fractional bee colony algorithms.

We are starting with the work of Chou et al. [69], which presents a fractional order particle swarm optimization to improve the classification of heart diseases via an XGBoost classifier. Furthermore [70] presented an improved fractional particle swarm optimization algorithm and shows its applicability to optimize support vector machine and K-Means algorithms. As used in the same domain, the researchers used a dataset where the task is to classify heart diseases.

An extension on the regular particle swarm optimization is the Darwinian particle swarm optimization algorithm, introduced in 2005 by Tillet et al. [71]. Couceiro et al. [72] introduced a fractional version of this algorithm. Ghamisi et al. later used these ideas in combination with random forest for classifying spectral images [73]. It is used in [74] again for spectral images, i.e., for the multilevel segmentation of spectral images later to be processed by support vector machines. Of course, we could also list these tasks in preprocessing, but since particle swarm is a classical optimization technique, we decided to use them in this category, i.e., optimization. Another application is discussed by Wang et al. in [75], where the researchers used a fractional Darwinian particle swarm optimization algorithm to optimize the feature selection for extreme learning machines. Another example to use an optimization procedure for preprocessing is found in [76], where the researchers used a fractional order Darwinian particle swarm algorithm to delineate cancer lesions. In a later step, these CT scan images are classified using decision trees. In both [77,78], the fractional-order Darwinian particle swarm is employed to segment MRI brain scans, which are then analyzed for strokes using random forest and support vector machines. Two further biomedical applications for fractional order Darwinian particle swarm are found in [79,80]. In both publications, the researchers used the discussed optimization algorithm to segment medical images, i.e., for detecting brain tumors and foot ulcers, respectively. Both publications then employ a supervised ML algorithm for classification, whereas the former uses Naive Bayes and a tree classifier, and the latter a fuzzy c-means classifier.

A different approach for optimizing machine learning approaches was taken in [81], where the researchers predicted the water quality using a fractional order artificial bee colony algorithm and decision trees. Another example for combining fractional order artificial bee colony algorithms and ML is given in [82] for predicting the nonperforming loans using a nonlinear regression model.

## 6. Discussion

Here, we summarize and discuss the performed literature analysis. We found a total of 60 publications that fit the criteria defined in Section 2. In the previous section, we separated all publications into three categories, i.e., preprocessing (Section 5.1), optimization (Section 5.3), and machine learning and fractional dynamics. We further sub-sectioned preprocessing and optimization because we found a variety of publications targeting similar problems in each of those categories. For preprocessing, we found three prominent types of applications, i.e., spectroscopy, biomedical and engineering applications. Contrary to that, we sorted all publications fitting into optimization into two categories. In contrast to preprocessing, we did not choose the categories with respect to the application but regarding the employed technique, whether it was a classic but then fractionalized gradient-based optimization technique or a gradient-free method.

A table containing all publications from Section 5 is given in Appendix A.

### 6.1. Preprocessing

The results for preprocessing show that a majority of the analyzed publications fall into the realm of spectroscopy. Here, researchers employed fractional derivatives to alter spectral data at hand to improve the machine learning algorithm’s accuracy. Overall, all of these approaches have in common that there is one or more *good* orders of fractional derivatives that enhance the data-based learning approach. We want to highlight that tree-based classifiers, e.g., random forest and XGBoost, are very common and perform well for these applications. We want to point out that Ref. [42] used fractional derivatives to preprocess hyperspectral data, which are then used to estimate the total nitrogen content in water via different machine learning algorithms, e.g., random forest.

Regarding biomedical applications, we find that here, similarly to preprocessing spectral data, researchers used fractional derivatives to preprocess their medical images, as done in [51] for CT images. Another example is to preprocess EEG or ECG signals, used to classify diseases or malfunctions as done in [46,48].

When it comes to engineering applications, we again found similar ideas, e.g., preprocessing images and spectral data to identify the malfunctions of solar panels or detect the status of construction equipment as performed in [53,54].

Overall, fractional derivatives can improve the accuracy of machine learning algorithms when employed for preprocessing signals, images, or spectral data. Furthermore, we find that there is a *sweet spot*, i.e., one or more orders of fractional derivatives that can significantly improve the accuracy of the employed algorithm. We further want to point out that, although fractional order preprocessing enhances the accuracy of machine learning applications, it also introduces a new hyperparameter, the order of the fractional derivative. Thus, it does not simplify the task at hand but instead provides a slight increase in accuracy at the cost of optimizing another hyperparameter.

Furthermore, we found that every application in which an actual fractional derivative was used for preprocessing involved the Grünwald–Letnikov fractional derivative.

Furthermore, we considered both data manipulation and feature extraction as preprocessing. Here, one can differ between manipulating the original data using a fractional derivative (e.g., [42]) and extracting features, i.e., error energy and parameters from the original data (e.g., [48]).

The authors believed that the preprocessing step using fractional derivatives is especially useful for non-neural network machine learning algorithms. The reason for this is that, given a deep neural network, the shallow layers of a neural network can make preprocessing obsolete as a neural network is capable of learning a huge variety of data manipulations. Thus, we suggest future research to test this hypothesis. Specifically, one needs to test whether a neural network can perform any preprocessing based on fractional derivatives.

Finally, an addition noteworthy to this discussion on fractional derivatives and preprocessing is the fractional measures of (signal) complexity, which can be used as a preprocessing step for machine learning applications. This is performed in [83], in which the researchers employed a fractional entropy measure to improve the fault detection for train doors.

### 6.2. Machine Learning and Fractional Dynamics

Regarding possible combinations of fractional dynamics and machine learning, we find that machine learning can be used to identify fractional order behavior from data, as performed in [55] or [59]. Overall, this is the category where we found the least amount of publications. This might be due to the fact that the fractional order modeling of dynamics requires profound knowledge in both mathematics and machine learning. Furthermore, we want to emphasize this category for its potential in the future. We suggest that researchers use, e.g., fractional orthogonal kernels in kernel-based machine learning algorithms as, e.g., support vector machines or Gaussian process regression to model systems from data [84].

### 6.3. Optimization

Regarding optimization, we split our review into the two main sections, i.e., gradient-based and gradient-free methods. The gradient-based optimization techniques feature gradient descent approaches where the integer-order gradient is replaced with a fractional-order gradient. For the gradient-free techniques, the idea is to replace, e.g., the velocity of a particle swarm algorithm with a fractional velocity.

Here, we want to point out two exemplary applications by Hapsari et al. [64,65] on well-known test-datasets, i.e., the Iris and the rainfall dataset, for the gradient-based optimization of machine learning algorithms.

For the gradient-free optimization algorithms, we highlight the work performed by Li et al. [70], where the researchers present an improved fractional particle swarm optimization algorithm to improve a support vector machine and k-means-based classification of heart diseases.

### 6.4. Bringing It All Together

This review’s motivation and the initial assumption is that combined approaches of fractional derivatives and machine learning improve machine learning tasks. We identified a total of three categories from the literature within our frame of keywords and criteria. The applications include processing, optimizing, and modeling fractional dynamics via machine learning. We discussed all of our findings in the previous sections. This section, however, brings all our results together and puts them into a bigger context. Here, we are taking into account the findings of the work of Niu et al. from 2021 [85], where the researchers proposed a triangle consisting of machine learning, fractional calculus, and the renormalization group. We depicted this triangle in Figure 1. The triangle suggests overlaps and further shows where potential combined approaches of machine learning and fractional calculus can be beneficial. Furthermore, we consider this triangle a closed loop; thus, future research might employ all of the mentioned tools, i.e., machine learning, fractional calculus, and the renormalization group, to tackle complex problems and describe complex systems. Thus, we describe our findings in this context and consider the results of another article by Niu et al. [86], entitled “*Why Do Big Data and Machine Learning Entail the Fractional Dynamics?*” in which the researchers further discuss ideas on the overlap between machine learning and fractional calculus, specifically for big data.

The following paragraphs discuss the overlaps shown by Niu et al., as depicted in Figure 1, compare them to the results of our literature review, and further provide evidence for a link between fractional calculus, machine learning, and the renormalization group.

#### 6.4.1. Optimization

We start by discussing the most obvious of the mentioned overlaps/connections, i.e., optimization. Fractional derivatives and, subsequently, fractional calculus is a powerful framework that can enhance optimization tasks, as fractionalized optimization algorithms are less prone to getting stuck in local minima/optima [87]. Thus, it is an upgrade to standard integer-derivative-based optimization tools for optimizing machine learning models’ hyperparameters. These optimization tools are also valuable for preprocessing. Here, we want to mention [77,78], these are exemplary applications in which this is used to segment MRI brain scans, later to be applied as an input for a machine learning algorithm to detect strokes.

#### 6.4.2. Variability

Given the results of our literature review, we can link several publications and the corresponding approaches to data variability, thus, providing evidence that fractional calculus and machine learning combined are tools capable of dealing with variability in data.

In [22], for example, the researchers employed fractional derivatives to preprocess Vis-NIR spectral data later to be used in a machine learning task to estimate organic matter content in arid soil. Similar preprocessing approaches are collected in Section 5.1.1. Thus, all of these articles deal with the spatial variability of the obtained spectral data. Furthermore, we can interpret these approaches as coping with climate variability as well, as they take into account or describe the components and interactions of the corresponding climate system.

We also find several publications where fractional derivatives and machine learning can deal with heart rate variability. This means that, in both [47,48], the researchers are using ideas related to fractional derivatives to, e.g., classify arrhythmic ECG signal. This also refers to human variability in general. We also refer to the work by Mucha et al. [50], where the researchers classify Parkinson’s disease based on online handwriting data, which were preprocessed using fractional derivatives.

According to the work of Niu et al. [86], variability is the central aspect of big data where employing fractional calculus and machine learning can be beneficial. Thus, we recommend considering approaches combining fractional derivatives and machine learning when dealing with variability in data and further recommend considering these for big data approaches.As the topic of big data would unnecessarily complicate this discussion, we refer the interested reader to [86,88].

#### 6.4.3. Nonlocal Models

Section 5.2 provides evidence for applying fractional calculus together with machine learning for nonlocal models. This means that all models based on fractional calculus are considered to be nonlocal because of the derivatives’ inherent *memory*. Here, we want to point out the work by [55], which provides a machine learning framework for identifying linear space-fractional differential equations from data.

#### 6.4.4. Renormalization Group

The renormalization group refers to a formal apparatus capable of dealing with problems involving many length scales. The basic idea is to tackle the problems in steps, i.e., one step at each length scale. For critical phenomena, the strategy is to carry out statistical averages over thermal fluctuations on all size scales. Thus, the renormalization group approach integrates all the fluctuations in sequence. This means starting on an atomic scale and moving step-by-step to larger scales until fluctuations on all scales are averaged out. The reason for doing so is that theorists have difficulties describing phenomena that involve many coupled degrees of freedom. For example, it takes many variables to characterize a turbulent flow or the state of a fluid near the critical point. Here, Kenneth and Wilson found the self-similarity of dynamic systems near critical points, which can be described using the renormalization group. Thus, the renormalization group is considered a powerful tool for studying the scaling properties of physical phenomena through its treatment of the scaling properties of complex and chaotic dynamics [89,90].

Thus, we want to discuss our findings in a bigger context together with renormalization group methods. We list and discuss evidence from our literature review for the proposed overlaps between fractional derivatives and machine learning with the renormalization group. Thus, can we find evidence for the keywords depicted in Figure 1? This means that the scaling law, complexity, nonlinear dynamics, fractal statistics, mutual information, feature extraction, and locality.

Again, we start our discussion with the most obvious, i.e., feature extraction. Regarding the triangle in Figure 1 [85], the researchers argued that, e.g., deep neural networks and their inherent feature extraction process, i.e., feeding and abstracting the input data from one layer to the next, learn to ignore irrelevant features while keeping relevant ones, which increases the efficiency of the learning process. The researchers further argue that this is similar to renormalization group methods, where only the relevant features of a physical system are extracted by integrating over short-scaled degrees of freedom to describe the system at larger scales. Their assumption is based on the findings of [91], which provides a mapping between the variational renormalization group and deep learning, whereas these findings are based on the Ising model. Furthermore, they discuss that the renormalization group is usually applied to physical systems with many symmetries. Contrary to that, deep learning is typically applied to data with a limited structure. Thus, the connection between the renormalization group and, in this case, deep neural networks, might not be generalizable to machine learning applications at large. Nonetheless, when talking about feature extraction, one can argue that fractional derivatives do just that in, e.g., [47,48]. These articles describe the feature extraction such that new parameters, e.g., the error energy of a signal when being modeled using a fractional transfer function, are found. This means that smaller-scaled features, i.e., the dataset itself, are not fed into the machine learning algorithm. Furthermore, we recommend using the fractional measures of complexity as another fractional calculus-based technique for feature extraction [83].

The researchers [85] discussed locality as the similarity of grouping spins in the black-spin renormalization group. Furthermore, they pointed out that this is similar to the shallow layers of a deep neural network in which the neurons are locally connected, e.g., without strong bonds, as described in [92]. Though this applies to deep neural networks, we did not find evidence in the reviewed publications. The reason here is that fractional derivatives behave nonlocally, i.e., spatio-temporal fractional derivatives have a certain degree of memory in both space and time.

When dealing with complexity, we need to consider that there is no unified definition for complexity yet. However, the researchers [85,90] argued that the overlap between fractional calculus and renormalization group methods might give us this definition and provide a framework to deal with complexity and thus answer the following two questions:How can we characterize complexity?What method should be used to analyze complexity to better understand real-world phenomena?

We consider finding answers to these questions idealistic. Thus, we need to weaken the statement to “*partially* answer this questions”. With much certainty, even when combined, fractional calculus, the renormalization group, and machine learning will only partially answer these questions, primarily because there are many different perspectives on the complexity of, e.g., data. Nonetheless, this triangle of powerful tools might expand definitions and find some definitions and characterizations for particular problems. To give an (in the authors’ opinion) reasonable example: this triangle might provide insights into the capability of machine learning to predict multifractal, chaotic, or stochastic time series data by characterizing a datasets memory and which time series can or cannot be learned by different machine learning algorithms.

Nonetheless, did we find evidence for complexity in our list of publications? We indeed found that there are phenomena that are better described using fractional calculus rather than integer-valued calculus, as described in [55]. Further in this work, the researchers used fractional calculus and machine learning to describe, e.g., financial time series data, e.g., the S&P500 index, which is widely accepted to be a complex phenomenon [93,94,95]. Furthermore, in [96], the researchers used renormalization group methods for analyzing the S&P 500 index. Thus, we conclude that we found evidence for this connection. Additionally, in [93], the researchers also found evidence for two universal scaling models/laws in finance. Thus, we found a connection and a potential research area for all three of the discussed techniques, i.e., the renormalization group, fractional calculus, and machine learning in financial time series analysis. Nonetheless, the discussed work of Gulian et al. [55] deals with linear space fractional differential equations and thus does not provide the framework to deal with nonlinear dynamics.

However, nonlinear dynamics (and subsequently chaos) are discussed in [82], where a fractional bee colony optimization algorithm is used to find the optimal parameters for a model which describes the nonlinear behavior of nonperforming loans. We also need to mention the work performed by Wang et al. [59], where fractional models for finance are parametrized using Gaussian process regression and predicted using recurrent neural networks. Thus, given this evidence, we suggest similar approaches to employ the renormalization group methodology in future research.

However, we did not find the applications of mutual information in this review. In [85], the researchers referred to the work by Koch-Janusz et al. [97] for the applicability of mutual information for both machine learning and the renormalization group. The researchers present a machine learning procedure to perform renormalization group tasks, thus, the reported connection. Here, the mutual information is used as an objective or target function where one of the employed neural networks aims to maximize the mutual information between two random distributions to deal with the reduction in degrees of freedom as the renormalization group would do. However, we did not find applications of mutual information for fractional derivatives and machine learning in our literature review, so we recommend future research to discuss the topic. For example, to what degree can machine learning algorithms be used to perform renormalization group tasks? Furthermore, are there non-model applications where these ideas can be used?

Furthermore, we did not find approaches dealing with scaling laws in the featured list of publications. Scaling laws, also known as power laws, are a description of the scale invariance of natural phenomena. Some notable examples according to [90] are, e.g., Zipf’s inverse power law which describes the relative frequency of word usage within a language [98].

Furthermore, finally, we also did not find an application of fractal statistics or fractals. However, given the numerous evidence for fractal behavior and statistics in various fields, e.g., in hydrology [99], rock mechanics [100] or finance [101], we still aim to provide a link. Here, we consider the previous discussion on the potential applicability of the triangle of machine learning, fractional calculus, and renormalization group on financial data. Again, we found evidence for fractal statistics present in economic data, i.e., the S&P 500 index [101]. Thus, we recommend future research for analyzing stock market data to employ ideas from fractional calculus, machine learning, and the renormalization group and subsequently to analyze the fractal behavior of these datasets. Furthermore, we believe that these ideas might be capable of analyzing, learning, and predicting a variety of complex real-life datasets, e.g., environmental data.

### 6.5. Problems

The last part of this discussion is to name two problems we encountered whilst conducting this literature review. This section should serve as a guideline on what to avoid and how to improve the future research field.

**Sometimes the fractional derivative is not discussed sufficiently:** we found literature that used fractional derivatives without stating the exact approach, e.g., [40]. However, as stated and referenced in [34], we assumed that the Grünwald–Letnikov fractional derivative is the employed fractional derivative.

We thus urge researchers to be specific about the employed fractional derivative, the corresponding discretization, and/or the employed software package to improve the transparency and reproducibility of their work. We further point out that hardly any article discussing the combined applications of machine learning and fractional derivatives listed in this review shows the employed discretization.

**Incorrect usage of keywords:** Some researchers assigned the term *fractional calculus* and/or *fractional derivatives* to their articles and describe the fractional calculus framework. However, in the end, they used a curve-fitting approach with fractional polynomial exponents and no fractional derivatives and/or calculus.

## 7. Conclusions

This review analyzes the combined applications of fractional derivatives and supervised non-neural network approaches. Our research shows that supervised machine learning and fractional derivatives are valuable tools that can be combined to, e.g., improve a machine learning algorithm’s accuracy. We found a total of three types of combined applications, i.e., preprocessing, modeling fractional dynamics via machine learning, and optimization.

We further found that most preprocessing applications are spectroscopical applications where a fractional derivative of a specific order is applied to the spectral data to improve the accuracy of the employed algorithm.

For optimization, we found two categories, gradient-based and gradient-free optimization algorithms. For gradient-based optimization, one can replace an integer-order gradient with a fractional-order gradient, thus obtaining a fractional gradient descent optimization algorithm. For the gradient-free optimization, the velocity in, e.g., a particle swarm algorithm, can be replaced by a fractional-derivative-based velocity. Hereby, one obtains a fractional order particle swarm optimization algorithm.

For the third type of combined applications, i.e., using a machine learning to model fractional dynamics, we found that machine learning and fractional derivatives, and subsequently fractional calculus are both techniques that deal with complex real-life phenomena, addressing the problem of complexity from different angles. Furthermore, this field improves kernel-based machine learning algorithms, such as Gaussian process regression and support vector machines, by providing kernels based on fractional derivatives to introduce memory into the modeling approach.

When bringing all our results together, we find a third tool conceptually linked to both, i.e., the renormalization group, which is a powerful approach for dealing with scaling laws, complex phenomena, complex dynamics, and chaos [102]. Thus, these three techniques, machine learning fractional calculus, and the renormalization group, form a triangle as proposed by Niu et al. [85]. Our review provides evidence for the proposed keywords, the corresponding connections, and thus the established links between machine learning, fractional calculus and the renormalization group.

In the author’s opinion, it will still take some time until the renormalization group will be part of mainstream computer science in combination with machine learning and fractional calculus. However, it is important to make today’s researchers, especially machine learning practitioners, aware and familiar with these topics as (supervised) machine learning is becoming popular in science in general. This means that, wherever there are available data, scientists are eager to find modern machine learning approaches to analyze, describe, and predict complex phenomena directly from data. Thus, machine learning is changing science overall [103,104].

Furthermore, as pointed out in [105], machine learning is sometimes considered an oracle that can learn any task from observations, i.e., predict chemical reactions and particle physics experiments, but not necessarily giving the practitioner scientist a deeper understanding of the process at hand. Thus, one might ask how and whether machine learning can contribute to our scientific understanding. The researchers further point out the multidisciplinary aspect of machine learning in general and that it will, because of this, undoubtedly keep increasing in popularity and can increase our scientific understanding of the studied processes. Thus, we propose that fractional calculus and the renormalization group will provide powerful tools in this context. Together with machine learning, these tools are able to bridge gaps between separated disciplines by improving numerical analysis, providing insights into complex phenomena, and by unveiling the hidden (or macroscopic; in the case of the renormalization group) mechanics governing these phenomena.

Regarding future research combining fractional derivatives and machine learning, we recommend testing whether neural networks can perform any kind of data manipulation that fractional derivatives can perform. This might be useful as it might avoid pointless contributions on the topic, i.e., fractional-derivative-based preprocessing for neural networks.

Furthermore, future research combining fractional derivatives, the renormalization group, and machine learning might give valuable insights into the learnability of machine learning algorithms and the corresponding underlying dynamics of the data. Specifically, one could connect ideas from the renormalization to a fractional derivative preprocessing while analyzing the performance of the employed machine learning algorithm. This means that finding the sweet spot of mandatory degrees of freedom and the corresponding order of fractional derivatives might provide insights about the memory of the studied data/system. Thus, providing the “minimal” representation of the data/system by optimal performance.

## Figures and Tables

**Figure 1 entropy-25-00035-f001:**
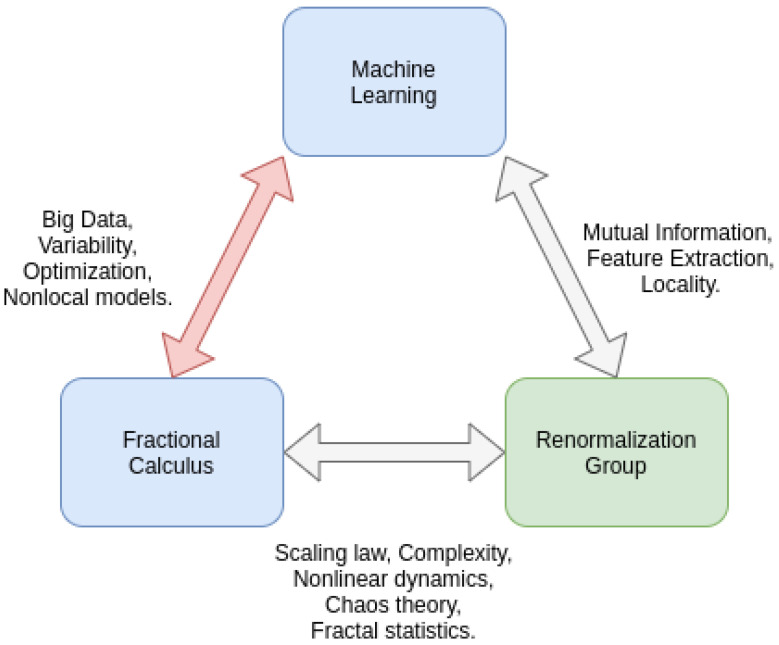
The triangle of machine learning, fractional calculus and the renormalization group as introduced by [85]. The current article focuses on the connection between machine learning and fractional calculus, i.e., the red double-arrow connecting the two blue boxes. This triangle is taken from [85], but we adapted it to better fit this review.

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
