# Peer review of "Combining Fractional Derivatives and Machine Learning: A Review"

_entropy, 2022, doi:10.3390/e25010035_

Round 1
Reviewer 1 Report
Entropy-2116457
“Combining Fractional Derivatives and Supervised Machine Learning: A Review”
Fractional calculus gained a lot of attention in the last couple of years. Researchers discovered that processes in various fields follow rather fractional dynamics than ordinary integer ordered dynamics, meaning the corresponding differential equations feature non-integer valued derivatives. There are several arguments for why this is the case, one of them being that fractional derivatives’ inherit spatiotemporal memory and/or the ability to express complex naturally occurring phenomena. Another popular topic nowadays is machine learning, i.e., learning behavior and patterns from historical data. In our ever-changing world with ever-increasing amounts of data, machine learning is a powerful tool for data analysis, problem-solving, modeling, and prediction. It further provides many insights and discoveries in various scientific disciplines. As these two modern-day topics provide a lot of potential for combined approaches to describe complex dynamics, this article reviews combined approaches of fractional derivatives and machine learning from the past, puts them into context, and thus provides a list of possible combined approaches and the corresponding techniques. Note, however, that this article does not deal with neural networks, as there already is profound literature on neural networks and fractional calculus. We sorted past combined approaches from the literature into three categories, i.e., preprocessing, machine learning & fractional dynamics, and optimization. The contributions of fractional derivatives to machine learning are manifold as they provide powerful preprocessing and feature augmentation techniques, can improve physically informed machine learning, and are capable of improving hyperparameter optimization. Thus, this article serves to motivate researchers dealing with data-based problems, to be specific machine learning practitioners, to adopt new tools and enhance their existing approaches.
This review analyzes the combined applications of fractional derivatives and supervised non-neural network approaches. The authors found a total of three types of combined applications, i.e., preprocessing, modeling fractional dynamics via machine learning, and optimization. Moreover, the authors found that most preprocessing applications are spectroscopical applications. Also, they found two categories with a fractional-order gradient Their review provides evidence for the established links between machine learning, fractional calculus and the renormalization group. In their opinion, future research combining fractional derivatives, the renormalization group, and machine learning might give valuable insights into the learnability of machine learning algorithms and the corresponding underlying dynamics of the data.
Comments
· This article aims to find overlaps between machine learning and fractional derivatives.
· The authors should present at least three recent refences that discussed the combined approaches of neural networks and fractional calculus.
· The authors present some effective combined applications of machine learning and fractional derivatives
· Section “2. Methodology” can be reduced and merged with the introduction section
· Ref [15] cab be replaced by de Oliveira, E.C.; Tenreiro Machado, J.A. A Review of Definitions for Fractional Derivatives and Integral. Mathematical Problems in Engineering 2014, 2014, 238459. doi:10.1155/2014/238459.
· Good effort is done in Sections 5 and 6 in classification of publications into three main categories on how to combine machine learning and fractional derivatives and its relations to a recent published paper “A New Triangle: Fractional Calculus, Renormalization Group, and Machine Learning”
· A brave act from authors that they list problems and drawbacks you faced through their research such as
- Sometimes the fractional derivative is not discussed sufficiently
- Incorrect usage of keywords:
- Grammar may be improved as possible
The article is well written, and it is a relevant contribution for the journal, so I find it appropriate to be published in Entropy. However, taking into account the above-mentioned remarks will improve the manuscript before final acceptance.
Reviewer 2 Report
This is a very useful review that analyzes the combined applications of fractional derivatives and supervised non-neural network approaches. The research showed that supervised machine learning and fractional derivatives are valuable tools that can be combined to, e.g., improve a machine learning algorithm’s accuracy. Three types of combined applications, i.e., preprocessing, modeling fractional dynamics via machine learning, and optimization are discussed.
I shall have no hesitation in recommending the paper for publication .
My only concern is that some journal papers e.g.,32,34,35,36 in the References must to be quoted by small letters. Is 33 a book or a paper?
